# The Use of Computational Approaches to Design Nanodelivery Systems

**DOI:** 10.3390/nano15171354

**Published:** 2025-09-03

**Authors:** Abedalrahman Abughalia, Mairead Flynn, Paul F. A. Clarke, Darren Fayne, Oliviero L. Gobbo

**Affiliations:** 1School of Pharmacy and Pharmaceutical Sciences, Trinity College Dublin, D02 PN40 Dublin, Ireland; abughala@tcd.ie (A.A.); flynnm27@tcd.ie (M.F.); 2Molecular Design Group, School of Chemical Sciences, Dublin City University, D09 V209 Dublin, Ireland; paul.clarke@uia.no; 3Department of ICT, University of Agder, 4879 Grimstad, Norway; 4DCU Life Sciences Institute, Dublin City University, Glasnevin, D09 V209 Dublin, Ireland; 5Trinity St. James’s Cancer Institute, St James’s Hospital, D08 NHY1 Dublin, Ireland

**Keywords:** molecular dynamics, artificial intelligence, machine learning, gold nanoparticles, lipid nanoparticles, nanomedicines

## Abstract

Nano-based drug delivery systems present a promising approach to improve the efficacy and safety of therapeutics by enabling targeted drug transport and controlled release. In parallel, computational approaches—particularly Molecular Dynamics (MD) simulations and Artificial Intelligence (AI)—have emerged as transformative tools to accelerate nanocarrier design and optimise their properties. MD simulations provide atomic-to-mesoscale insights into nanoparticle interactions with biological membranes, elucidating how factors such as surface charge density, ligand functionalisation and nanoparticle size affect cellular uptake and stability. Complementing MD simulations, AI-driven models accelerate the discovery of lipid-based nanoparticle formulations by analysing vast chemical datasets and predicting optimal structures for gene delivery and vaccine development. By harnessing these computational approaches, researchers can rapidly refine nanoparticle composition to improve biocompatibility, reduce toxicity and achieve more precise drug targeting. This review synthesises key advances in MD simulations and AI for two leading nanoparticle platforms (gold and lipid nanoparticles) and highlights their role in enhancing therapeutic performance. We evaluate how in silico models guide experimental validation, inform rational design strategies and ultimately streamline the transition from bench to bedside. Finally, we address key challenges such as data scarcity and complex in vivo dynamics and propose future directions for integrating computational insights into next generation nanodelivery systems.

## 1. Introduction

The process of developing new therapeutic drugs is both costly and time-consuming, often exceeding a decade, with research and development (R&D) costs ranging from USD 1 billion to over USD 2 billion per drug [1]. Despite these significant investments, the overall success rate remains low. For instance, a recent analysis of nearly 4000 drug candidates from the US, EU and Japan indicates that only approximately 12.8% of compounds entering clinical trials eventually receive regulatory approval [2]. This low success rate, combined with prolonged development timelines, significantly delays access to life-saving therapies for patients with unmet medical needs [3].

Compounding these challenges, traditional drug discovery heavily relies on animal models for in vivo testing, a practice increasingly criticised for ethical concerns and its limited ability to predict human response. These limitations have prompted the global adoption of the 3Rs framework (Reduction, Refinement, Replacement) to minimise animal use [4]. To address these challenges, computational approaches have emerged as solutions, to accelerate drug development while reducing reliance on animal testing.

Nanomedicine, as defined by the European Medicines Agency (EMA) is “the application of nanotechnology in view of making a medical diagnosis or treating or preventing diseases” [5]. Nanotechnology broadly refers to the intentional design, characterisation and production of materials, structures, devices and systems at the nanoscale (1–100 nm), although some definitions extend beyond this range [6]. These nanomaterials possess unique physicochemical properties that enable applications in diagnostics and therapeutics [7] and as nano-based drug delivery systems (NDDSs). By engineering nanocarriers with tunable size, shape and surface chemistry, researchers can improve tissue permeability, reduce toxicity and enhance drug efficacy [8].

Among the diverse range of nanomaterials—including lipids, polymers, dendrimers and metallic nanoparticles such as silver and gold (Figure 1) [9]—gold nanoparticles (AuNPs) and lipid nanoparticles (LNPs) have garnered significant interest due to their distinct advantages. AuNPs exhibit unique optical properties, chemical stability and tunable surface chemistry [10], enabling their application in photothermal therapy, imaging and drug delivery [11]. LNPs, on the other hand, have attracted considerable attention since the FDA’s 1995 approval of Doxil^®^ [12], the first nanodrug utilising PEGylated liposomal doxorubicin for cancer therapy. This breakthrough was followed by the 2018 approval of Onpattro^®^ (Patisiran), the first RNA interference (RNAi) therapeutic targeting hereditary transthyretin-mediated amyloidosis [13]. The versatility of LNPs was further exemplified during the COVID-19 pandemic, where their ability to encapsulate nucleic acids was pivotal in the development of mRNA vaccines, including Pfizer-BioNTech’s Comirnaty and Moderna’s Spikevax^®^.

Despite these advances, the widespread clinical application of nanomedicine remains limited [14]. A major challenge is the incomplete understanding of how nanomaterial properties—such as size, shape and surface chemistry—translate to biological outcomes in the human body [15]. In recent years, computational approaches have emerged as powerful tools to tackle these challenges. MD simulations and Artificial Intelligence (AI) are among the most promising computational techniques for modelling nanocarrier interactions in biological environments [16]. MD simulations offer a “computational microscope” to observe atomic-level interactions and stability of nanocarriers in silico [17], while AI techniques can mine larger datasets to identify patterns and optimise formulations rapidly [18].

This review is unique, as it highlights the recent integration of computer-aided development, AI and nanomedicine in the design of nanodrugs—an emerging approach that is transforming drug discovery and delivery. However, given that both nanomedicine and computational simulations are relatively new fields, few comprehensive reviews effectively integrate these domains. To address this gap, the present review focuses on MD simulations in the design of AuNPs and LNPs for drug delivery, as well as on AI applications in LNP design. We begin by discussing the foundational principles of MD simulations and then critically evaluate how these computational techniques can be leveraged to enhance NDDSs, comparing their respective strengths and limitations. Next, we examine the use of AI in designing LNPs, highlighting how data-driven approaches can streamline and optimise formulation strategies. Finally, we discuss the current challenges that limit the practical adoption of these tools and explore prospective directions for future research.

**Figure 1 nanomaterials-15-01354-f001:**
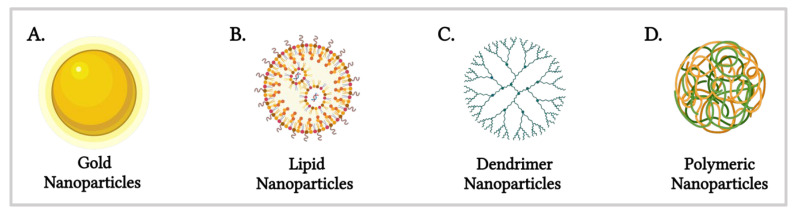
Schematic representation of various nanomaterials utilised in nano-based drug delivery systems (Created in BioRender. Gobbo, O. (2025) https://BioRender.com/bx4wrz1 [19]).

## 2. Molecular Dynamics Simulations for the Development of Nano-Based Drug Delivery

### 2.1. Fundamentals of Molecular Dynamics Simulations

MD simulations, first pioneered by Alder and Wainwright in 1957 [20], are computational techniques that model the behaviour of molecules and atoms by numerically solving Newton’s equations of motion using predefined force fields [21]. MD simulations provide critical insights into nanoparticle stability, membrane interactions and drug loading efficiency by capturing the atomic-scale behaviour of nanomaterials [22,23]. These simulations can be conducted at varying levels of resolution, each with distinct advantages and limitations.

All-atom MD (AAMD) simulations explicitly represent each atom, offering highly detailed molecular insights into molecular interactions and physiological processes [22]. However, due to their computational expense, AAMD simulations are typically restricted to shorter timescales and smaller system sizes [22]. In contrast, coarse-grained MD (CGMD) simulations reduce computational complexity by grouping clusters of atoms into simplified representations known as “beads”, thereby enabling simulations of larger biomolecular assemblies over longer timescales [24]. These clusters of atoms are given generalised physicochemical character and joined together with “CG bonds”. How atoms are clustered and transformed into these “beads” is determined by a mapping operation, *M*, where the configuration, *R*, of the CG model is a function of the configuration, *r*, of an original atomistic model. The cartesian coordinates, *R_I_*, of bead *I* are determined as a linear combination of atomic Cartesian coordinates, *r_i_*, with constant, positive coefficients that often correspond to, e.g., the centre of mass or geometry for the associated atomic group (Equation (1)) [25].(1)$$R_{I}=M_{I}\left(r\right)=\sum_{i}c_{Ii}r_{i},$$

Different CGMD methods are defined by two components: (1) the mapping operator and (2) defining the interactions between CG “beads” or sites. Different mapping operators are often based upon the chemical understanding and needs of the researcher. In the case of proteins, for example, atoms are often clustered by their associated amino acid group, where one site is representative of that amino acid (Figure 2B). By positioning this site at the α-carbon of the amino acid, this allows for detailed reconstruction of the protein backbone and detection of secondary structures through Ramachandran maps [26,27]. Despite its simplicity, this chemically informed method has been crucial to elucidating principles of protein folding and interactions [28,29].

Different interactions between CG sites should capture the effects of different atomistic details that have been eliminated from the CG model. While being highly efficient, CG methods should retain “correct physics” for fundamental insight and accurate predictions. Many ways of defining these interactions have been proposed. Foundational methods often follow a “top-down” or “bottom-up” approach [25].

A “bottom-up” CG model is constructed based on the atomistic model for the same system. For this, a statistical mechanics-based framework is often applied in which the many-body potential of mean force (PMF) sits as the central quality. The many-body PMF, *W*, is completely specified by the underlying atomistic model and the CG mapping and is truly a potential that generates mean forces. Methods of approximating and utilising *W* are covered in more detail by dedicated reviews [30].

A “top-down” model is generally constructed without the consideration of a more detailed parent model. They often use real experimental observations, physicochemical intuition and low-parameter phenomenological models. For example, chemically specific top-down models often employ interaction potentials with simple functional forms that are parametrised to reproduce thermodynamic properties, where sites correspond to 3–4 heavy atoms. The popular Martini model extends this paradigm by providing transferable potentials that describe the effects of hydrophobic, van der Waals and electrostatic interactions between sites as a function of their polarity and charge [30].

All cases of CGMD simulations sacrifice some atomistic detail, yet remain highly effective for studying large-scale nanoparticle behaviour, long-term stability and membrane interactions (Figure 2). Additionally, when higher resolution is required, reverse mapping techniques can reconstruct atomistic details from coarse-grained models [24].

A typical MD workflow includes selecting the starting structure, preparing the simulation system, running the simulation on high-performance computing (HPC) resources and analysing trajectories to extract molecular properties such as system stability and binding energies. The most frequently used MD simulation software includes AMBER, CHARMM, GROMACS and LAMMPS [31].

To highlight the complementary strengths of different modelling approaches, Table 1 provides a comparative summary of all-atom MD, CGMD and emerging specialised tools such as DockSurf [32], with a focus on their resolution, advantages, limitations and typical applications in nanodelivery research.

### 2.2. Molecular Dynamics Simulations for Designing Gold Nanoparticles

#### 2.2.1. Simulations on Optimal Size and Surface Charge Density of AuNPs

Surface charge density (SCD) can significantly influence the interaction of AuNPs with biological membranes, particularly their penetration, permeability and toxicity [33]. By utilising CGMD simulations, Lin et al. [34] demonstrated that the interaction of AuNPs with lipid membranes is highly dependent on SCD, influencing both cellular uptake and cytotoxicity. Their simulations revealed that cationic AuNPs exhibit strong adhesion to negatively charged membranes, facilitating penetration at moderate SCD levels of up to 50% without causing immediate structural damage. However, beyond this threshold, significant membrane disruption occurs, compromising membrane integrity and potentially leading to cytotoxic effects.

Similarly, Quan et al. [35] explored the influence of SCD on asymmetric membranes, which more closely resemble those found in mammalian cells. Their simulations revealed that cationic AuNPs with an SCD of up to 70% exhibited enhanced penetration, facilitating cellular uptake. However, beyond this threshold, increased charge density led to membrane disruption, characterised by flip-flop and loss of membrane asymmetry, which could compromise structural integrity and potentially hinder further uptake (Figure 3). Their CGMD simulations further suggest that low-SCD AuNPs are better suited for drug delivery systems due to their reduced cytotoxicity, whereas highly charged AuNPs, capable of significant membrane disruption, may be advantageous for tumour-selective therapies.

With AuNPs typically ranging from 1 to 100 nm, determining the optimal size for specific biomedical applications is crucial. Size directly influences the optical and physicochemical properties of AuNPs, affecting tissue permeability and the interactions with cell membranes [36]. Gupta et al. [37] employed CGMD simulations to examine how AuNP size influences penetration through the skin’s lipid bilayer. Their findings revealed that smaller AuNPs penetrate the bilayer with minimal disruption, whereas larger AuNPs induced membrane disruption and formed hydrophobic cavities. Notably, the study observed a self-healing effect, where the bilayer reorganised and restored its structural integrity after AuNP translocation. This suggests that AuNP-induced disruptions may be temporary, potentially affecting membrane permeability and drug retention within tissues.

Furthermore, Gupta et al. [38] employed CGMD simulations to investigate the combined effects of AuNP surface charge and size on skin permeability. Their findings revealed that both cationic and anionic AuNPs predominantly remained adsorbed at the lipid bilayer headgroup and did not penetrate. In contrast, neutral hydrophobic AuNPs penetrated the bilayer within approximately 200 nanoseconds. Moreover, smaller neutral hydrophobic AuNPs induced greater membrane disruption, whereas larger particles exhibited reduced permeability. These MD simulations offer molecular-level insights to guide the design of transdermal drug delivery systems.

#### 2.2.2. Simulations on Stability of AuNPs

The small size and high surface energy of AuNPs make them prone to agglomeration in solvents. To prevent agglomeration, an effective monolayer must be formed, making ligand capping density a key factor in nanoparticle stability [39]. Researchers have employed CGMD simulations and AAMD simulations to examine how nanoparticle size, capping length and capping density influence monolayer formation and stability.

For instance, Colangelo et al. [40] explored the relationship between peptide structure and its arrangement on AuNPs using AAMD simulations and experimental techniques such as FTIR spectroscopy. Their findings demonstrated that higher capping densities resulted in compact and well-ordered monolayers, whereas lower capping densities led to disordered peptide organisation. Additionally, longer peptides exhibited greater structural organisation, forming extended β-sheet domains, particularly on larger AuNPs. This research underscores the role of ligand density in monolayer stability and nanoparticle functionalisation.

Nqayi et al. [41] investigated the effects of varying AuNP size, polyethylene glycol (PEG) length and ligand density on nanoparticle stability using CGMD simulations. Their findings indicate that smaller AuNPs have lower coordination numbers, leading to increased reactivity and reduced stability. In contrast, larger AuNPs have higher coordination numbers and lower surface energy, enhancing stability. The study also found that PEG molecules with a chain length of n = 2 provide optimal stability, while longer chains do not significantly enhance stability due to steric hindrance. Additionally, electron-donating groups (–NH_2_, OH) enhance nanoparticle stability, whereas electron-withdrawing groups (COOH) increase reactivity, leading to reduced stability. These findings emphasise the need for precise optimisation of AuNP size and surface functionalisation to enhance stability for drug delivery.

#### 2.2.3. Interaction of AuNPs with Biological Membranes

Upon in vivo administration, nanoparticles interact with biological fluids, forming a protein corona which influences their biological activity [42]. This adsorption process can induce structural modifications in proteins, potentially causing denaturation, conformational changes and functional loss [43]. Gaining insights into the composition and dynamics of protein coronas is essential for engineering nanoparticles with improved safety and functionality in biomedical applications.

Sajib et al. [43] employed CGMD simulations and AAMD simulations to investigate the formation and structural properties of the protein corona on bare AuNPs, a key factor influencing their interaction with biological membranes. Their findings revealed that nanoparticle size plays a critical role in protein adsorption and orientation, directly affecting AuNP-membrane interactions. Smaller AuNPs formed a stable, single-layer corona, while larger AuNPs led to multilayered protein adsorption, driven by stronger protein–protein interactions (Figure 4). Further analysis showed that adsorption onto AuNPs induced structural changes in smaller proteins, with a loss of α-helices and an increase in disordered conformation. While a well-formed protein corona can improve biocompatibility and enhance drug delivery, excessive protein adsorption may trigger opsonisation, accelerating nanoparticle clearance and reducing bioavailability. These findings emphasise the role of MD simulations in predicting corona stability and guiding the design of AuNP-based drug delivery systems. Notably, these results align with Zhang et al. [44], who also observed that AuNPs induce structural changes in adsorbed proteins, with size-dependent effects influencing aggregation behaviour.

Beyond computational studies, experimental validation is essential to confirm these computational findings under physiological conditions. Using circular dichroism (CD) spectroscopy and MD simulations, Kaumbekova et al. [45] investigated the conformational stability of bovine serum albumin (BSA) upon AuNP interaction. Their results revealed that 5 nm AuNPs alone caused minimal structural changes, but in the presence of NaCl, a synergistic destabilisation effect led to partial α-helix loss. Further MD simulations showed that smaller AuNPs induced greater conformational changes, consistent with earlier computational predictions.

MD simulations can also provide valuable insights into the potential adverse effects of inorganic nanoparticles arising from their interactions with biomolecules. Shao et al. [46] employed AAMD simulations to investigate the allosteric effects of AuNP binding on human serum albumin (HSA), a key transport protein in the bloodstream. Their findings revealed that AuNP binding induced conformational changes in approximately 10% of HSA residues, not only at the adsorption site but also at key ligand-binding regions such as fatty acid, thyroxin and metal ion sites. These structural modifications may influence HSA’s biological function, highlighting the need to consider allosteric effects in nanoparticle design. This study demonstrates how MD simulation can aid in developing functionalised AuNPs to reduce unintended protein interactions.

Tavanti et al. [47] used CGMD simulations to investigate how common blood proteins—serum albumin, haemoglobin, complement C3 and α1-antiproteinase—interact with AuNPs capped with hydrophobic ligands. Their simulations revealed that protein binding was primarily driven by hydrophobic interactions between amino acid residues and the ligand, leading to the formation of a stable protein corona. Despite strong adsorption, the secondary and tertiary structures of the bound proteins remained largely intact. These findings suggest that hydrophobic capping ligands could be utilised in nanodelivery system design to help maintain protein structural integrity and minimise unintended biological interactions.

Beyond traditional MD and CGMD simulations, recent advances have introduced specialised tools that directly address protein adsorption onto inorganic surfaces. One example is DockSurf, a molecular modelling software developed to predict protein orientations and adsorption modes on Au{111} surfaces [32].

Unlike standard MD, which is often sensitive to the initial placement of proteins, DockSurf systematically rotates proteins through different orientations and evaluates their stability using energy maps, quickly identifying the most favourable protein–surface configurations. These predictions have been benchmarked against MD trajectories for common serum proteins such as albumin, demonstrating their reliability in capturing how proteins organise at the nanoparticle interface. Such interface-focused modelling is particularly useful for nanodelivery applications, where the mode of protein attachment can determine whether a nanoparticle circulates safely in the body, avoids premature clearance, and successfully delivers its therapeutic payload.

Consistent with this, recent studies emphasise that engineering the nanoparticle-protein interface through ligand design, e.g., by using polyethylene glycol or zwitterionic coatings, can modulate corona formation, cellular uptake and therapeutic activity [48].

### 2.3. Molecular Dynamics Simulations for Designing Lipid Nanoparticles

#### 2.3.1. Molecular Insights into Lipid Nanoparticles Interactions with Biological Membranes

Traditional liposomes have been investigated for nucleic acid delivery; however, their low encapsulation efficiency and limited delivery success present significant challenges [49]. To overcome these limitations, cationic lipids were introduced due to their ability to form electrostatic complexes with negatively charged nucleic acids, thereby enhancing encapsulation efficiency and facilitating intracellular delivery [50].

Using CGMD simulations, Ou et al. [51] investigated how variations in core hydrophobicity and lipid coatings of cationic lipid nanoparticles (cLNPs) influence their membrane specificity. Their findings revealed that highly unsaturated cationic lipid coatings significantly enhanced the membrane-binding probability of cLNPs, particularly toward bacterial cell membranes. This interaction followed a two-step mechanism: electrostatic adhesion, followed by hydrophobic insertion, leading to membrane disruption and bacterial toxicity. Interestingly, these cLNPs did not bind to red blood cell membranes, highlighting their selective antibacterial activity (Figure 5). These insights provide a foundation for the rational design of LNPs with improved bacterial targeting while minimising off-target effects.

#### 2.3.2. MD Simulation in Mucoadhesive Nanocarrier Design

Beyond bacterial targeting, MD simulations have also been instrumental in screening ligand–membrane interactions and optimising nanoparticle adhesion for ocular drug delivery. This was demonstrated by Pai et al. [52], who used MD simulations to evaluate the mucoadhesive properties of three ligands—chitosan oligosaccharide (COS), stearylamine (STA) and cetrimonium bromide (CTAB)—on Mucin-4 (MUC4), a glycoprotein abundant in the ocular epithelium. Their results revealed that COS exhibited the highest mucoadhesion, forming simultaneous multiple interactions, including hydrogen bonding, ionic interactions and hydrophobic interactions. STA showed moderate binding, while CTAB demonstrated minimal interaction with MUC4. These findings were validated through in vitro and in vivo studies in rats. While MD simulations effectively predicted ligand binding, the model did not account for physiological factors such as tear clearance, which influence drug retention. Nevertheless, these findings highlight the potential of computational approaches in guiding the rational design of mucoadhesive nanocarriers for enhanced ocular drug delivery.

#### 2.3.3. MD Simulations for Optimising Drug Permeability

MD simulations provide a bottom-up approach to designing drug delivery systems by identifying molecular factors that contribute to poor bioavailability. By predicting drug-membrane interactions, MD complements experimental findings and aids in selecting optimal ligands to enhance permeability, stability and therapeutic efficacy. A key application of MD simulations in drug delivery is its ability to identify molecular interactions that hinder drug permeability, guiding strategies to improve drug transport. This was exemplified by Li et al. [53], who employed MD simulations to investigate the transmembrane characteristics and low ocular bioavailability of the active drug tetrandrine (TET). The simulations revealed that TET’s hydrophobic groups strongly interacted with the POPC lipid membrane tails, while its two amine groups in the hydrophilic region hindered its passage through the membrane centre. These findings, consistent with experimental results, guided researchers toward functionalising TET-loaded LNPs with cationic ligands to improve ocular permeability and bioavailability.

A similar study by Li et al. [54] utilised MD simulations to examine the interactions of baicalein (BAI) with a simulated POPC membrane. The simulations revealed that BAI exhibited strong hydrophilic interactions with the phosphate group of the membrane, while its weak hydrophobic interactions with the lipid tails contributed to its low ocular availability. These findings highlighted the barriers to membrane permeability, which limit BAI’s effectiveness as an ocular drug. Although MD simulations did not explicitly test BAI’s interactions with LNPs, they provided key insights into its poor transmembrane permeability, guiding the development of trimethyl chitosan coated LNPs to enhance bioavailability.

#### 2.3.4. Molecular Dynamics Simulations for Optimising Lipid Nanoparticles Assembly and Stability

MD simulations can also be employed in assessing the self-assembly and stability of LNPs, providing molecular-level insights into their structural organisation. Fernandez-Luengo et al. [17] employed CGMD simulations using the MARTINI force field to investigate the self-assembly and surfactant behaviour of LNPs composed of tripalmitin lipids with Tween 20 as a stabilising agent. Their findings revealed that tripalmitin LNPs exhibit high lipid mobility, with a liquid-like core rather than a rigid, well-ordered structure. Additionally, the study assessed the role of Tween 20 in stabilising LNPs. The simulations demonstrated that Tween 20 did not form a homogeneous monolayer around the LNP but instead assembled into uneven patches, leaving some regions of the nanoparticle surface exposed. This irregular distribution of the surfactant could impact the steric stabilisation of LNPs and their interactions with biological environments. These findings highlight the ability of MD simulations to predict nanocarrier stability and optimise lipid–surfactant compositions, reducing reliance on trial-and-error experimental approaches in LNP formulation.

Long-term physical stability remains a major challenge in the development of LNPs, especially those containing ionisable amino lipids. Gindy et al. [55] identified Ostwald ripening, a process in which smaller nanoparticles dissolve and redeposit onto larger ones, as a key factor contributing to LNPs instability. Using CGMD simulations, they investigated how different phospholipid compositions—1,2-dimyristoyl-sn-glycero-3-phosphocholine (DMPC), Distearoylphosphatidylcholine (DSPC) and dilauroylphosphatidylcholine (DLPC)—affected nanoparticle stability. Their findings revealed that LNPs formulated with DMPC exhibited more uniform molecular packing, which significantly slowed down Ostwald ripening compared to LNPs containing DSPC and DLPC. These computational predictions were further validated by experimental studies, which demonstrated that DMPC-based LNPs maintained improved structural stability over extended storage periods. This study highlights the critical role of MD simulations in predicting optimal lipid compositions stabilising LNPs.

LNPs have become essential for delivering genetic materials, such as siRNA and mRNA, in therapeutic and vaccine applications. However, their stability is highly dependent on environmental factors, including the presence of ethanol, a common solvent used during LNP formulation. Hardianto et al. [56] investigated the molecular effects of ethanol using MD simulations. Their findings demonstrated that ethanol disrupted lipid packing, increased solvent penetration and compromised the encapsulated genetic material. Root mean square deviation (RMSD) analysis revealed progressive structural instability, while solvent-accessible surface area (SASA) calculations showed enhanced siRNA exposure, suggesting a reduction in protective lipid coverage. Additionally, hydrogen bond analysis indicated that ethanol formed interactions with siRNA, interfering with lipid-RNA bonding networks and accelerated destabilisation. These results emphasise the need for rapid ethanol removal during LNP production to preserve nanoparticle stability and ensure effective drug delivery.

Paloncyová et al. [57] investigated the structural organisation and stability of RNA-loaded LNPs across different pH condition using MD simulations. The LNP formulation examined included ionisable lipids (ILs), cholesterol, DPPC and PEGylated lipids, closely resembling those in mRNA vaccines such as Pfizer-BioNTech’s COVID-19 vaccine. The simulations revealed that at low pH, ILs remained protonated, forming stable electrostatic interactions with RNA and maintaining efficient encapsulation within the LNP core. However, as pH increased and ILs become deprotonated, these interactions weakened, leading to RNA displacement and structural rearrangement of the LNP.

These findings suggest that optimising IL composition and charge behaviour could enhance RNA stability and delivery efficiency. Importantly, atomistic MD simulations verified these observations, reinforcing the predictive capability of computational methods in guiding rational LNP design.

## 3. Artificial Intelligence and Machine Learning for Designing Lipid Nanoparticles

### 3.1. Fundamentals of Artificial Intelligence and Machine Learning

Artificial Intelligence (AI) comprises computational methods that simulate human intelligence, allowing machines to perform tasks such as learning, reasoning and problem solving [58]. In nanomedicine, AI facilitates the analysis of complex datasets, aiding in the design and optimisation of nanoparticles for drug delivery. AI algorithms can predict nanoparticle behaviour in biological environments, improving targeting specificity and reducing off-target effects [59].

Machine Learning (ML), a branch of AI, involves training algorithms on large datasets to recognise patterns and make data-driven decision [60]. In nanodelivery systems, ML models can optimise parameters such as nanoparticle size, surface chemistry and drug release kinetics [59]. This predictive capability accelerates the development of nanoparticles with desired properties, streamlining the drug delivery process (Figure 6).

A major turning point in machine learning was the emergence of deep learning, which enables algorithms to automatically learn patterns from complex data. Unlike earlier approaches that depended on manually engineered features, deep learning can extract informative representations directly from raw inputs such as images, text or molecular structures. This breakthrough, highlighted in a review by LeCun et al. [61], transformed areas such as image and speech recognition and opened new directions in fields including genomics and drug discovery. These advances now underpin applications in nanomedicine, where learning from high-dimensional biological datasets is essential for predicting nanoparticle behaviour and drug release profiles, as well as patient-specific responses.

Traditionally, the development of designing LNPs relied on extensive trial and error, consuming significant time and resources. However, AI-driven methods have transformed this process by enabling rapid computational screening of vast lipid libraries, predicting optimal nanoparticle formulations prior to laboratory testing. Building on these capabilities, the next section of our review examines how AI-driven models optimise LNP design for drug delivery, with a focus on mRNA vaccines and gene therapy applications. While AuNPs have been studied for their tunable properties, the application of AI in their design remains limited.

### 3.2. Current Applications of Artificial Intelligence to Design Lipid Nanoparticles

#### 3.2.1. Applications of Artificial Intelligence to Design Lipid Nanoparticles for mRNA Delivery

The design and formulation of nanomaterials have greatly benefited from AI, which allows researchers to screen thousands of material combinations before laboratory formulation [18]. LNPs have emerged as the leading non-viral delivery system of mRNA, playing a pivotal role in gene therapy and vaccine development. A critical component of LNPs, ionisable lipids, has traditionally been developed through experimental screening or rational design-methods that are often labour-intensive and may overlook optimal lipid structures. However, advances in AI, particularly deep learning, have revolutionised this field by enabling the rapid and efficient design of ionisable lipids, subject to the availability of good quality, relevant and large datasets.

The AI-Guided Ionisable Lipid Engineering (AGILE) platform developed by Xu et al. [62] is a deep learning-powered approach, designed to accelerate the development of ionisable lipids for mRNA delivery [18]. AGILE employs a graph neural network (GNN), a machine learning model that processes molecular structures as graphs, to predict how well different LNPs will deliver mRNA into cells. The system follows a two-phase learning strategy: self-supervised pre-training on a virtual library of 60,000 lipids, followed by supervised fine-tuning using data from 1200 experimentally synthesised lipids, ultimately screening a 12,000-candidate lipid library (Figure 7). This approach enabled accurate predictions of mRNA transfection potency (mTP), identifying high-performance candidates such as H9 and R6. H9 LNPs demonstrated a 7.8-fold increase in mRNA delivery to muscle tissue with reduced off-target accumulation, while R6 LNPs exhibited a five-fold increase in transfection efficiency compared to H9, highlighting AGILE’s ability to identify ionisable lipids optimised for macrophage-targeted mRNA delivery. However, despite AGILE’s ability to overcome data scarcity through pre-training and fine-tuning, it does not account for data imbalance, which contributes to significant prediction errors [63].

To address AGILE’s limitations, Wu et al. [63] developed TransLNP, a transformer-based deep learning model designed to enhance the screening and selection of LNPs for mRNA delivery, analysing the relationship between lipid molecular structure and transfection efficiency. Given the challenge of imbalanced LNP datasets, they introduced the BalMol block, which balances the data by adjusting how frequently different lipid types are represented in the dataset, leading to more reliable predictions. By integrating TransLNP with the BalMol block, they reduced the mean squared error (MSE) to 1.47 on the AGILE dataset, significantly improving predictive accuracy. Their study confirms that AI, particularly ML, can be effectively employed in LNP design, facilitating quicker and more precise predictions for mRNA therapeutic applications, thereby advancing the development of effective drug delivery systems.

While AGILE and TransLNP primarily focus on LNP screening and transfection prediction, Bae et al. [64] used a Random Forest (RF) regression model to analyse 213 LNP formulations, incorporating 314 molecular features to predict mRNA delivery efficiency. Their study identified phenolic hydroxyl groups within ionisable lipids as a key factor in enhancing mRNA encapsulation and expression. Additionally, they examined the impact of carbon chain length, finding that longer chains led to unstable multi-compartmental structures, ultimately reducing delivery efficiency. The RF model also demonstrated strong predictive power, achieving a Pearson correlation coefficient of 0.845, indicating its ability to accurately capture the relationship between molecular structure and mRNA expression.

#### 3.2.2. Applications of Artificial Intelligence to Design Lipid Nanoparticles for Gene Therapy

Despite the success of mRNA vaccines and hepatic RNA delivery, targeted LNPs are needed to expand RNA-based therapies for genetic disease [65]. Lung-targeted gene therapy remains challenging but holds promise for treating conditions such as cystic fibrosis [66] and chronic obstructive pulmonary disease [67]. To address these challenges, Witten et al. [57] developed Lipid Optimisation using Neural Networks (LiON), an AI-driven framework designed to enhance ionisable LNP formulations for gene therapy. Using directed message-passing neural networks (D-MPNNs), a subset of deep learning, LiON analyses the structures of ionisable lipids and predicts their ability to deliver nucleic acids efficiently. To train the model, the authors compiled a dataset comprising over 9000 LNP activity measurements, including in vitro and in vivo studies.

LiON evaluated 1.6 million lipid structures in silico and successfully identified two novel lipid structures, FO-32 and FO-35, which demonstrated highly efficient mRNA delivery to the lungs, muscles and nasal tissue. FO-32 exhibited comparable efficiency to leading nebulised mRNA delivery systems in the mouse lung, and both FO-32 and FO-35 efficiently delivered mRNA to ferret lungs.

## 4. Rational Engineering of Nanoparticle Interfaces and Implications for Drug Delivery

The data reviewed in this work underscore that the performance of nano-based drug delivery systems is strongly influenced by the rational engineering of their interfaces: the dynamic region where a nanoparticle first encounters the biological milieu. This bio–nano interface governs the earliest stages of interaction with cells, proteins and extracellular components, ultimately dictating biodistribution, clearance rates and therapeutic outcomes.

MD simulations have been particularly valuable for dissecting these relationships at atomic and mesoscale resolution. By systematically varying parameters such as surface charge density, ligand orientation, PEGylation extent and hydrophobicity, MD studies reveal how small adjustments can produce significant changes in membrane adhesion, penetration kinetics and protein corona composition [68]. For example, optimising AuNP surface charge within a defined threshold range can enhance membrane binding and uptake while minimising structural disruption and cytotoxicity [69]. Similarly, ligand length and capping density have been shown to influence steric repulsion, corona evolution and the likelihood of immune system recognition—factors that are directly relevant to circulation half-life and tissue targeting.

From a biological perspective, these engineered interface features have direct and measurable impacts on delivery performance [70]:Target tissue accumulation can be increased through receptor-specific ligand display and by reducing non-specific adhesion.Intracellular trafficking can be modulated to favour endocytic pathways that improve cytosolic delivery and endosomal escape.Drug release kinetics can be tailored via stimulus-responsive linkers that respond to pH, enzymatic activity or redox gradients within pathological microenvironments (e.g., acidic microenvironment around tumor cells).Immunogenicity and clearance can be reduced by controlling nanoparticle corona formation through antifouling chemistries and optimised PEG architectures.

AI complements this mechanistic insight by identifying complex, non-linear relationships between surface chemistry and biological performance that are often difficult to predict empirically. ML models trained on physicochemical–biological datasets can forecast how specific lipid blends or polymer architectures will influence both stability and therapeutic potency. When integrated with MD-derived mechanistic predictions, this approach supports a closed-loop design–test–refine strategy, enabling rapid iteration towards optimal interface designs without relying solely on trial and error [71].

The convergence of computational modelling and rational interface engineering therefore represents a powerful shift in nanomedicine development: from empirical formulation towards predictive, hypothesis-driven optimisation. This approach not only accelerates preclinical design but also improves the probability that in vitro gains will translate to in vivo efficacy by anticipating protein corona dynamics, variable pH environments and fluid shear stresses present in physiological systems. Ultimately, the deliberate design of nanoparticle interfaces, guided by both MD simulations and AI-driven analytics, offers a route to more predictable pharmacokinetics, enhanced therapeutic indices and reduced translational failure rates, thereby facilitating the advancement of safer and more effective patient-specific nanomedicines [72].

## 5. Conclusions

Computational methods, encompassing both MD simulations and AI are reshaping nano-based drug delivery systems by optimising formulations and providing deeper insights into nanoparticle–biological macromolecule interactions. However, computational predictions must be validated under physiologically relevant conditions to ensure their accuracy and applicability. Experimental studies, such as circular dichroism for protein structural changes, cryo-TEM for nanoparticle morphology, dynamic light scattering for stability, and in vitro/in vivo biodistribution assays, serve to confirm simulation-derived hypotheses, identify unanticipated behaviours and refine computational models.

The integration of these approaches supports a closed-loop design framework:Simulation-led screening to identify promising formulations.Experimental validation to confirm physicochemical properties and biological activity.Feedback to models to update simulation parameters based on empirical data, improving predictive accuracy.

This iterative cycle reduces reliance on trial-and-error experimentation, enhances the likelihood that in vitro gains will translate in vivo, and accelerates the progression from benchtop discovery to clinical application. We have added this integration framework to the concluding analysis to emphasise that simulation and experimental validation are not parallel alternatives but interdependent components of a unified NDDS design strategy.

MD simulations offer molecular-level precision, allowing researchers to predict how variations in size, surface charge and ligand density influence the stability and functionality of AuNPs and LNPs. Meanwhile, AI-driven algorithms complement these insights by rapidly screening vast chemical libraries, identifying optimal nanocarrier compositions and predicting drug delivery efficiency. Together, these computational approaches bridge the gap between theoretical design and experimental validation, significantly accelerating developing in nanomedicine.

By identifying formulations with enhanced biocompatibility and targeted delivery, computational approaches streamline therapeutic development while reducing reliance on labour-intensive trial-and-error experimentation. They also contribute to reducing reliance on animal models, aligning with ethical guidelines to promote refinement and eventual replacement of in vivo testing. Despite these advances, challenges remain, such as the need for standardised, high-quality datasets and models that more accurately reflect the complexity of human physiology. Addressing these gaps will be crucial for translating computational predictions in clinically viable nanomedicine solutions, ultimately enabling more precise and patient-specific therapies. As these technologies continue to evolve, their integration holds immense potential to transform patient care through more effective, targeted and personalised drug delivery systems.

## 6. Challenges and Future Directions

The integration of AI and MD simulations in designing nanodelivery systems presents significant potential but also several challenges. MD simulations are computationally intensive. Running large-scale simulations requires substantial processing power, often necessitating high-performance computing resources and parallel computing with Graphics Processing Units to improve efficiency. Without these resources, simulations are constrained to shorter timescales and smaller system sizes, limiting their ability to capture biologically relevant processes.

Additionally, the accuracy of MD simulations depends on the quality of force fields used to model atomic interactions. Current force fields may not always capture the complex behaviour of nanoparticles in biological environments, leading to discrepancies between computational predictions and experimental outcomes. Refining these force fields is important for improving the predictive accuracy of MD simulations and ensuring reliable results for drug delivery applications and more accurate modelling of nanoparticle-biological interactions.

Beyond system-level considerations, there are also critical interface-specific challenges. Protein and peptide adsorption onto nanoparticle surfaces is inherently dynamic, involving conformational rearrangements and competitive exchange within the protein corona. These processes occur on timescales that remain largely inaccessible to conventional MD or CGMD simulations. While coarse-grained models extend accessible timescales, they can oversimplify key details such as orientation-dependent binding or partial unfolding, both of which strongly influence nanoparticle recognition and biological fate. Standard MD is also constrained by its sensitivity to the initial placement of proteins, which may bias adsorption outcomes and reduce reproducibility.

New approaches such as DockSurf mitigate this by systematically exploring orientation landscapes to provide unbiased predictions of protein–surface geometries [32]. In parallel, experimental and computational studies have shown that ligand chemistry is a dominant factor shaping corona composition and dynamics, with direct consequences for cellular uptake, immune recognition, and therapeutic efficacy [48]. Collectively, these limitations highlight the need for hybrid frameworks that integrate atomistic accuracy with efficient sampling, and that are closely coupled with experimental validation, to capture the full complexity of nano–bio interfaces.

AI-driven approaches also face several key challenges. One major limitation is the availability and quality of data. AI models, particularly deep learning systems, require large datasets to make accurate predictions. However, in nanomedicine, experimental data remains scarce and often lacks standardisation. Another challenge is the interpretability of AI-driven predictions. Many machine learning models, particularly deep learning algorithms, function as “black boxes,” meaning that their decision-making processes are difficult to understand [73]. This lack of transparency poses obstacles for clinical adoption, as regulatory bodies and healthcare professionals require explainable models to ensure patient safety and therapeutic reliability.

Another promising avenue is the integration of AI with physics-based simulations, such as MD. Hybrid models that combine AI’s data-driven insights with the mechanistic accuracy of MD simulations can improve nanoparticle design by offering a comprehensive understanding of their behaviour in biological environments [74]. Furthermore, the development of Explainable AI techniques (XAI) is essential to enhance transparency in AI-driven predictions. By making AI models more interpretable, XAI can increase trust among researchers, clinicians and regulatory authorities, facilitating their adoption in nanomedicine [75].

Finally, personalised nanomedicine represents an exciting frontier. Leveraging AI to design patient-specific nanocarriers could revolutionise treatment by considering genetic, environmental and lifestyle factors. This personalised approach could enhance therapeutic efficacy, minimise adverse effects and lead to more precise drug targeting [73].

## Figures and Tables

**Figure 2 nanomaterials-15-01354-f002:**
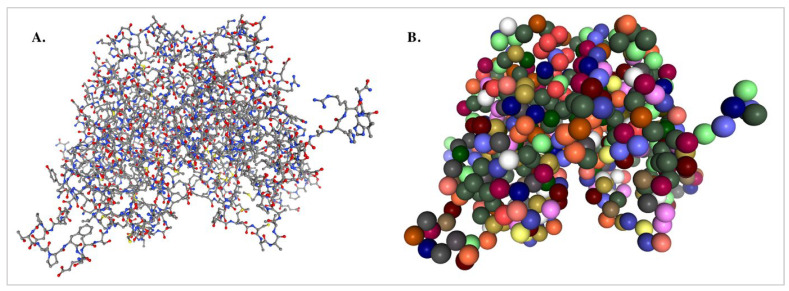
(**A**) Atomistic ball and stick rendering of the estrogen receptor α (PDB code: 3ERT) versus (**B**) coarse grained visualisation of the protein (Created using NGLView (version 2.7.7) [26]). Colour scheme in (**A**); grey: carbon, red: oxygen, yellow: sulphur; blue: nitrogen. In (**B**), the colours correspond to different amino acids.

**Figure 3 nanomaterials-15-01354-f003:**
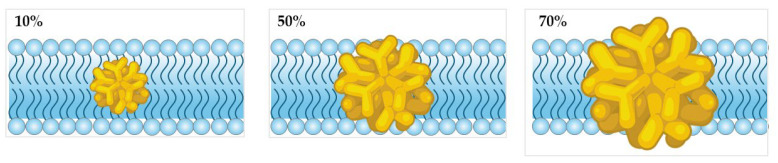
Effect of AuNP surface charge density (SCD) on lipid bilayer penetration, based on CGMD simulations. Higher SCD increases membrane interaction, leading to disruption beyond 70% SCD (Created in BioRender. Gobbo, O. (2025) https://BioRender.com/tocgzot [19]).

**Figure 4 nanomaterials-15-01354-f004:**
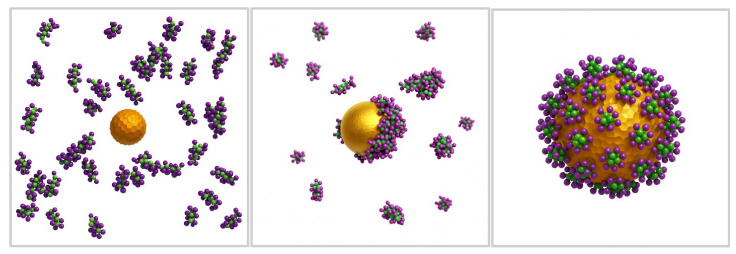
Molecular simulations of protein corona formation on AuNPs and their interactions with biological membranes. (Created using ChatGPT 4 and Adobe Photoshop 2025 Version 26.8 and Adapted from Sajib et al. [43]).

**Figure 5 nanomaterials-15-01354-f005:**
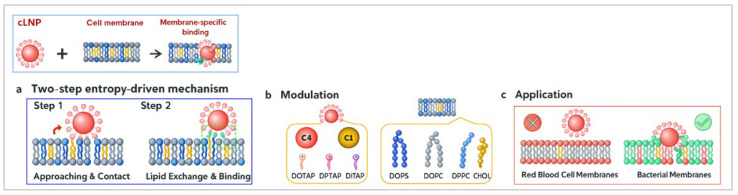
(**a**) cLNPs bind via an initial electrostatic adsorption to the bilayer, following by lipid exchange that drives hydrophobic locking. (**b**) The binding strength is tuned by the NP-Core hydrophobicity and coating-lipid tail unsaturation. (**c**) This enables selective adhesion to bacterial-like membranes over red blood cell models. (Created using ChatGPT 4 & Adobe Photoshop 2025 Version 26.8 and adapted from Ou et al. [51]).

**Figure 6 nanomaterials-15-01354-f006:**
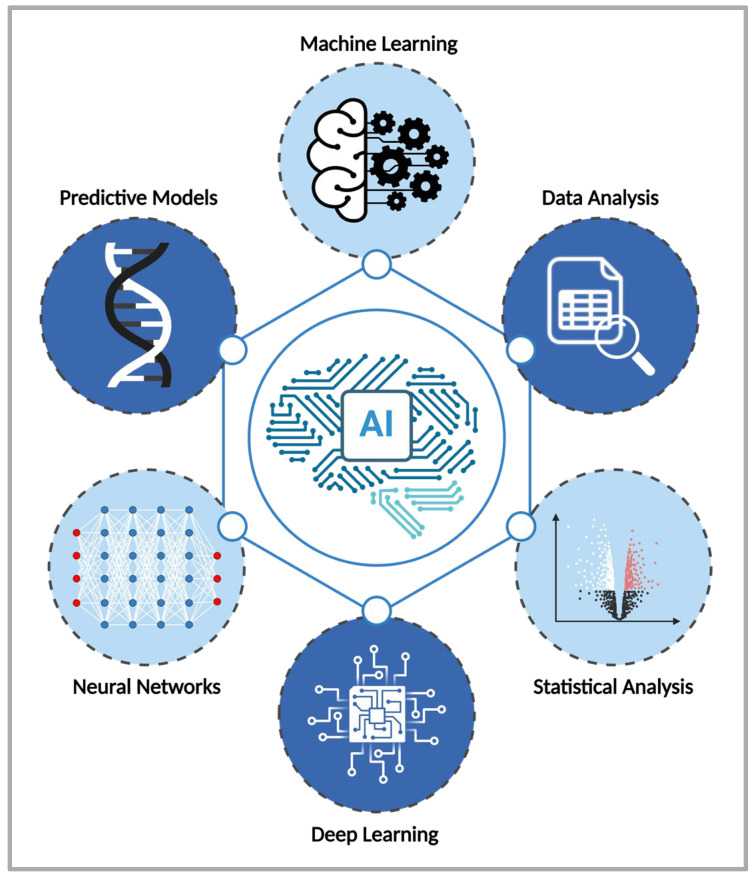
Key AI components and techniques used for the design and optimisation of LNPs (Created in BioRender. Gobbo, O. (2025) https://BioRender.com/gx3k1pf [19]).

**Figure 7 nanomaterials-15-01354-f007:**
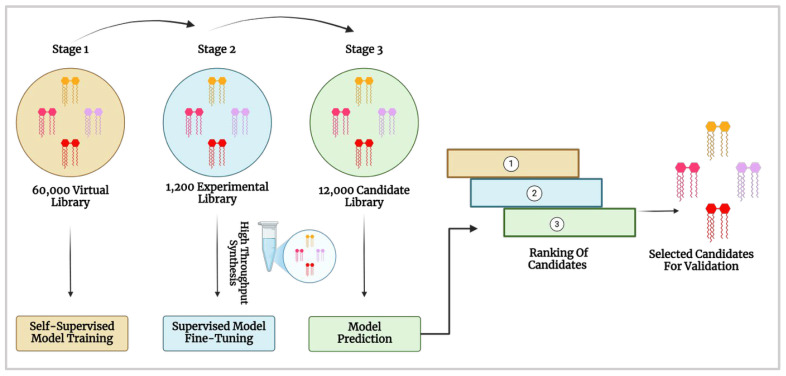
AI-driven screening and optimisation of ionisable lipids for mRNA delivery. The process involves self-supervised training on a virtual lipid library, fine-tuning with experimental data and model-based candidate ranking for validation (Created in BioRender. Gobbo, O. (2025) https://BioRender.com/g7h4ysc [19] and adapted from Xu et al. [62]).

**Table 1 nanomaterials-15-01354-t001:** Comparative overview of MD, CGMD and DockSurf for modelling nano–bio interfaces.

Method	Resolution/Scale	Advantages	Limitations
All-Atom MD	Atomistic Detail (Å; ns–µs).	Explicit representation of every atom. Highly accurate molecular interactions. Secondary structure detection.	Computationally intensive. Restricted to short timescale and small systems.
CGMD	Coarse-Grained (µs–ms).	Extends timescales and system sizes. Computationally efficient. Transferable force fields (e.g., Martini)	Sacrifices atomistic detail. May oversimplify orientation-dependent binding and unfolding.
DockSurf	Protein–Surface Docking.	Rapid exploration of protein adsorption. Unbiased by initial placement.	Limited to predefined surfaces.

## Data Availability

No new data were created or analysed in this study. Data sharing is not applicable to this article.

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
