# Peer review of "The Use of Computational Approaches to Design Nanodelivery Systems"

_nanomaterials, 2025, doi:10.3390/nano15171354_

Round 1
Reviewer 1 Report
Comments and Suggestions for Authors
Dear Editor, I carefully read the manuscript entitled “The Use of Computational Approaches To Design Nanodelivery Systems,”. The manuscript offers a comprehensive and timely review of computational approaches—particularly molecular dynamics (MD) simulations and artificial intelligence (AI)/machine learning—in the rational design of nanodrug delivery systems. The coverage of gold and lipid nanoparticle platforms is thorough and the review adeptly integrates recent progress in the application of in silico methods to advance therapeutic efficacy, safety, and translation to the clinic.
The overall quality is very good and I raise some suggestion of corrections. Here are my remarks:
1. A notable gap, however, is the underrepresentation of recent advances in the computational modeling of protein and peptide adsorption at inorganic and nanostructured surfaces, as well as the rational functionalization of nanoparticles for targeted delivery or bioactivity. This aspect is crucial given the review’s emphasis on interface engineering as a basis for optimizing nanocarrier performance, yet the current references do not reflect some of the state-of-the-art tools and exemplars in this area. To strengthen the review and offer a more complete perspective, I strongly recommend the inclusion and the discussion of the following references:
- Docksurf software, J. Chem. Inf. Model. 2023 (63): 5220−31. Another approach to explore protein/inorganic surface preditction.
- Engineering Nanoparticle–Protein Interfaces for Therapeutic Activity, Nanomaterials 2021 (11): 502-20. Example of how the mode of biomolecule attachment onto inorganic nanoparticles dramatically affects biological function.
- A classical reference in AI deeplearning. Nature 2015 (7553), 436–444.
2. Ensure discussion of current limitations and future perspectives in MD/CGMD covers not only system-scale but also interface-specific modeling (including protein and peptide adsorption phenomena).
3. Consider a brief summary or table contrasting available methods (general MD, CGMD, and specialized tools like DockSurf) for modeling nano-bio interfaces.
4. Enhance the discussion around the rational engineering of nanoparticle interfaces, with direct relevance to drug delivery performance and biological outcomes.
5. Demonstrate the necessary integration of both simulation and experimental validation in the design workflow of next-generation NDDS.
Reviewer 2 Report
Comments and Suggestions for Authors
Nanomaterials and computational approaches are two current hot research fields. The authors synthesized these two fields together through drug nanodelivery systems, specifically reviewing molecular dynamic simulation (MD) and artificial intelligence (AI) applications on gold nanoparticles (AuNPs) and lipid nanoparticles (LNP) systems. The authors briefly summarized the essences of MD and AI, then showed their recent contributions to simulations of Size and Surface Charge Density and interactions with biological membranes for AuNP through MD, and Designing Lipid Nanoparticles through MD and AI. The review is interesting; I recommend its publication in nanomaterials.
Some minor comments:
1.Since section 2.2.1 (line 136) and section 2.2.2 (179) have the same titles but refer to different tasks, sections 2.2.1 and 2.2.2 may be revised to “2.2.1 Simulations on Optimal Size and Surface Charge Density of AuNPs” and “2.2.2 Simulations on Stability of AuNPs”
- There are TWO Figure 6 (line 286 and line 410). The first one should be “Figure 5”.
3.Data Availability Statement section needs to be specified
- Acknowledgments section needs to be simplified.
Reviewer 3 Report
Comments and Suggestions for Authors
The presented review paper concerns the computer-aided design of drug carriers based on nanoparticles. Two selected methods are discussed, molecular dynamics and artificial intelligence/machine learning. This is a very interesting approach to drug design that can reduce costs and time.
Two nanoparticle types were mainly considered: gold nanoparticles and liposomes/lipid based macromolecules. The manuscript is well organized and the selection of discussed references is relevant. The Authors concentrate on Coarse-Grained Molecular Dynamics over All Atom simulations, but taking into account the size of studied systems it is reasonable approach. The section concerning AI/ML is relatively shorter, although this approach is new. The Authors could improve the manuscript by adding short discussion of other candidates for drug nanocarriers like fulerenes or carbon nanotubes and their derivatives. I would also suggest expanding on the proposed integration of MD simulations and AI/ML methods mentioned in section five.
Overall I recommend the publication of the manuscript after minor revision.
There are some minor issues that should be corrected, before the manuscript can be considered for publication:
1. Fig. 5 on page seven is mislabeled. The caption registers it as Fig. 6.
2. XAI abbreviation in section 5 was not introduced. It should be added in previous sentence.
3. The Acknowledgment section should be cleaned of the generic text from template.
Reviewer 4 Report
Comments and Suggestions for Authors
This is a short review discussing modelling approaches, mostly based on molecular dynamics, of nanosystems designed for drug delivery. AI driven methods are also discussed, and the authors stress the convergence of 'traditional' modelling approaches and AI-based methods for the development of comprehensive tools aimed at in-silico designing drug-delivering systems.
The review is clearly meant for an audience just entering the field of molecular simulations and as such can be less relevant for readers already experts in the usage and development of classical tools for macromolecular and supramolecular chemistry.
In particular, I believe that the introduction to coarse-grained (CG) methods (sec. 2) could be highlighted more, with at least a basic introduction to the physical and mathematical approximation involved, at least to make more aware a less experienced reader of the potentialities and pitfalls of CG approaches.
The presentation of AI methods (sec. 3) for describing drug delivery systems is interesting but still short and covering a very limited ensemble of cases and examples, probably closer to the authors' knowledge and experience. Again, it can serve as a preliminary introduction to the field, but in a very limited way.
In short, 1) I think that the paper could be considered a preliminary presentation of the vast scientific field it attempts to address.
2) This is of course an acceptable choice made by the authors, but is should stressed clearly in the introduction and conclusion sections
3) Basic papers related to the physico-mathematical definition of methods are missing (e.g. Espanol et al.).
As such, I would recommend some revision of the text, stating clearly the limitations in scope of the paper and if possible a short clear exposition of basic methods, with appropriate references.
Round 2
Reviewer 4 Report
Comments and Suggestions for Authors
The authors took into account most of the reviewers' comments and suggestions.